# Is liver transplantation 'out-of-hours' non-inferior to 'in-hours' transplantation? A retrospective analysis of the UK Transplant Registry

Neil Halliday,[1,2] Kate Martin,[3] David Collett,[3] Elisa Allen,[3] Douglas Thorburn[1]

¹Sheila Sherlock Liver Centre, Royal Free Hospital and the UCL Institute for Liver and Digestive Health, London, UK
²Institute of Immunity and Transplantation, University College London, London, UK
³Statistics and Clinical Studies, NHS Blood and Transplant, Bristol, UK

**Correspondence to**
Dr Neil Halliday;
neilhalliday@doctors.org.uk

## ABSTRACT

**Objectives** Increased morbidity and mortality have been associated with weekend and night-time clinical activity. We sought to compare the outcomes of liver transplantation (LT) between weekdays and weekends or night-time and day-time to determine if 'out-of-hours' LT has acceptable results compared with 'in-hours'.

**Design, setting and participants** We conducted a retrospective analysis of patient outcomes for all 8816 adult, liver-only transplants (2000–2014) from the UK Transplant Registry.

**Outcome measures** Outcome measures were graft failure (loss of the graft with or without death) and transplant failure (either graft failure or death with a functioning graft) at 30 days, 1 year and 3 years post-transplantation. The association of these outcomes with weekend versus weekday and day versus night transplantation were explored, following the construction of a risk-adjusted Cox regression model.

**Results** Similar patient and donor characteristics were observed between weekend and weekday transplantation. Unadjusted graft failure estimates were 5.7% at 30 days, 10.4% at 1 year and 14.6% at 3 years; transplant failure estimates were 7.9%, 15.3% and 21.3% respectively. A risk-adjusted Cox regression model demonstrated a significantly lower adjusted HR (95% CI) of transplant failure for weekend transplant of 0.77 (0.66 to 0.91) within 30 days, 0.86 (0.77 to 0.97) within 1 year, 0.89 (0.81 to 0.99) within 3 years and for graft failure of 0.81 (0.67 to 0.97) within 30 days. For patients without transplant failure within 30 days, there was no weekend effect on transplant failure. Neither night-time procurement nor transplantation were associated with an increased hazard of transplant or graft failure.

**Conclusions** Weekend and night-time LT outcomes were non-inferior to weekday or day-time transplantation, and we observed a possible small beneficial effect of weekend transplantation. The structure of LT services in the UK delivers acceptable outcomes 'out-of-hours' and may offer wider lessons for weekend working structures.

## INTRODUCTION

Increased morbidity and mortality have been observed with out-of-hours clinical practice in a range of settings[1] which has, in part, been ascribed to differing clinical service provision

### Strengths and limitations of this study

► This is the first study to address whether there is a weekend effect on clinical outcomes for liver transplantation in the UK.
► The study was based on an assessment of a large, unbiased, multicentre dataset of all UK liver transplant operations occurring in the study period.
► The UK Transplant Registry is a well curated, highly complete database enabling the generation of risk-adjusted models including recipient, donor and technical parameters that may have influenced outcomes.
► Transplantation settings offer the ability to explore 'out-of-hours' outcomes where the timing of clinical event is not determined by the recipient's clinical status.
► The major limitations of this study include the inability to identify causative factors, nor identify confounding factors that may be driving differences in outcomes.

through the week. Liver transplantation (LT) services are structured differently to most other clinical services[2] due to the complexity, time sensitivity, scarcity of donations and potential risk of LT. All aspects of LT care are consultant-led, with a standardised service provided at all times and multiple clinical teams including surgeons, anaesthetists, physicians, radiologists and intensive care specialists involved in each case, assisted by specialist co-ordinating staff. Whether this service structure protects against potential weekend effects has not previously been explored in the UK.

Several studies have reported excess mortality associated with weekend hospital admission in the UK[3–7] and elsewhere.[1 8 9] However, despite adverse weekend effects being observed in many studies, they are not consistent across all diagnoses or presentations and only a proportion of clinical presentations have an observable

weekend effect.[10–12] Even with conditions associated with adverse weekend effects, conflicting outcomes have been reported[13–15 16–19] and similarly to medical presentations, surgical and intensive care unit (ICU) studies have conflicting results.[9 10 20–32] These findings suggest that adverse weekend effects are complex, disease specific and may have different underlying causes including service structure.[19] Despite this, a recent assessment of the impact of enhanced 7-day working practices in the UK did not show a beneficial impact on adverse weekend outcomes.[33]

The current evidence for out-of-hours LT outcomes is mixed. No increased risk of mortality or graft failure was demonstrated with weekend or night-time LT in a multi-centre American study of nearly 95 000 transplants.[34] Another single-centre American study demonstrated no increase in surgical complications or long-term mortality but did show an increase in early mortality following LT at night.[35] Renal transplantation at the weekend was not associated with increased mortality or graft failure in a UK study of nearly 13 000 transplants,[36] a smaller German study[37] (although higher rates of surgical complications were observed) or a large American study.[38]

It remains unclear whether the observed excess mortality associated with weekend admission is a product of differing case severity,[39 40] admission thresholds,[41] systematic differences in care delivery, structure and staffing of services, quality of care, poor quality data recording[17] or is an artefact.[42] As weekend effects are specific to different diagnoses and clinical scenarios, if the differences in clinical outcomes are due to service provision and structure, each clinical service should be tested for acceptable of outcomes across the week.

The UK delivers LT services with a high volume, low centre-number model, with seven centres providing services for a population of approximately 65 million people, each performing between 30 and 172 deceased donor, adult-recipient LTs annually.[43] With the development of a national organ retrieval service with stipulated retrieval response times and increasing reliance on donation after cardiac death (DCD), the incidence of out-of-hours transplantation has been increasing. We wanted to establish whether the model of service delivery in the UK ensures consistency in outcomes throughout the day and week. We retrospectively assessed the hazard of graft failure or transplant failure following single organ LT across all UK centres, comparing weekday with weekend and day with night transplantation.

## METHODS

Data on all adult recipients (≥17 years) of liver-only transplants from deceased donors in the UK, under the National Health Service (NHS) between 1 January 2000 and 31 December 2014, were obtained from the UK Transplant Registry and followed up to 18 February 2016.

Night-time procurement was deemed to be any liver donation where the liver perfusion start time was between 19:00 hours and and 07:00 hours. We estimated transplant operation time by adding donor liver perfusion start time (effectively the time of organ retrieval) to cold ischaemia time (CIT) (the time between perfusion and the re-establishment of circulation to the graft, within the recipient). Liver perfusion data were not collected before 2000, and so we have only included transplants since 2000 in the cohort. Night-time transplantation was defined as operation time between 19:00 hours and 07:00 hours. Weekend transplantation was defined as any transplant operation time between 17:00 hours on a Friday and 08:00 hours on a Monday whereas weekday transplantation included all other time points. These time points were selected to ensure that our findings were comparable to other published studies.[34]

The primary outcomes were graft failure and transplant failure. Transplant failure was defined as the earlier of graft failure or patient death (graft failure before death, graft failure and death, or death with a functioning graft were classed as an event), whereas graft failure classed graft failure before death or graft failure and death as an event. Therefore, patients who underwent retransplantation would have had an event of graft failure associated with their first liver transplant.

Student's t-test, $\chi^2$ and log-rank tests were used to compare continuous, categorical and failure rate data, respectively. Cox proportional hazards models were built to estimate graft and transplant failure at 30 days, 1 year and 3 years post-transplant. HRs for different time periods were found by including a period factor in the model. Factors considered for inclusion in the model are listed in table 1. Stepwise variable selection, a combination of forwards and backwards selection, was used to identify factors to be included in the models for the different end points guided by a combination of statistical significance and clinical considerations.

Less than 5% of values for each baseline patient, donor and operative characteristic were missing (see online supplementary table A1). Missing values for the following recipient factors: international normalised ratio, sodium, creatinine and bilirubin (used in calculating model of end-stage liver disease (MELD) and UK model for end-stage liver disease (UKELD) score), body mass index (BMI), CIT, on renal support, acute failure grade, inpatient, ventilated, oesophageal varices, sepsis confirmed, previous abdominal surgery; and surgical factors: suboptimal organ appearance, night-time procurement, night-time transplant and weekend transplant were imputed using multiple imputation based on chained equations.[44] This involved generating 11 data sets with imputed values, with the median of continuous variables and the modal value of the categorical variables being used to produce the final data set. These factors were investigated for any pattern of missingness, but there was generally no evidence of systematic difference in missingness for transplant failure, with the exception of donor type. At 1-year follow-up, there were more cases of missing patient or graft outcomes for DCD transplants (14%) compared with donor with brainstem death (DBD) transplants (7.7%).

**Table 1** Characteristics of weekday and weekend liver-only transplants in the UK, 1 January 2000 to 31 December 2014

| | Weekday transplant | Weekend transplant | |
|---|---|---|---|
| | N (%) unless otherwise stated | | P value |
| **Total number of transplants** | **5613 (64)** | **3203 (36)** | |
| **Recipient characteristics** | | | |
| Transplant year | | | |
| *2000* | 312 (6) | 209 (7) | |
| *2001* | 345 (6) | 188 (6) | |
| *2002* | 335 (6) | 232 (7) | |
| *2003* | 288 (5) | 207 (6) | |
| *2004* | 340 (6) | 239 (7) | |
| *2005* | 315 (6) | 172 (5) | |
| *2006* | 321 (6) | 196 (6) | 0.002 |
| *2007* | 341 (6) | 198 (6) | |
| *2008* | 379 (7) | 195 (6) | |
| *2009* | 379 (7) | 187 (6) | |
| *2010* | 379 (7) | 212 (7) | |
| *2011* | 406 (7) | 209 (7) | |
| *2012* | 441 (8) | 227 (7) | |
| *2013* | 511 (9) | 246 (8) | |
| *2014* | 521 (9) | 286 (9) | |
| Super-urgent | 803 (14) | 523 (16) | 0.01 |
| Age at transplant, mean (SD) | 50 (13) | 50.2 (12) | 0.5 |
| Male gender | 3405 (61) | 1899 (59) | 0.2 |
| Caucasian | 4922 (88) | 2814 (88) | 0.8 |
| MELD at transplant, mean (SD) | 19.1 (9) | 19.5 (9) | 0.05 |
| UKELD at transplant, mean (SD) | 56.0 (6) | 56.4 (7) | 0.02 |
| Primary liver disease | | | |
| *Cancer* | 320 (6) | 171 (5) | |
| *Hepatitis C infection* | 746 (13) | 395 (12) | |
| *Alcohol related liver disease* | 1180 (21) | 626 (20) | |
| *Hepatitis B infection* | 153 (3) | 107 (3) | |
| *Primary Sclerosing Cholangitis* | 451 (8) | 276 (9) | 0.14 |
| *Primary Biliary Cholangitis* | 520 (9) | 333 (10) | |
| *Autoimmune hepatitis* | 453 (8) | 255 (8) | |
| *Metabolic liver disease* | 316 (6) | 172 (5) | |
| *Acute liver disease* | 647 (12) | 414 (13) | |
| *Retransplant* | 539 (10) | 304 (9) | |
| *Other* | 288 (5) | 150 (5) | |
| ABO blood group | | | |
| *O* | 2405 (43) | 1318 (41) | |
| *A* | 2338 (42) | 1341 (42) | 0.01 |
| *B* | 648 (12) | 372 (12) | |
| *AB* | 222 (4) | 172 (5) | |
| Renal support | 690 (12) | 421 (13) | 0.2 |
| Inpatient | 1585 (28) | 975 (30) | 0.03 |
| Ventilated | 585 (10) | 373 (12) | 0.08 |
| Oesophageal varices | 3406 (61) | 1991 (62) | 0.17 |
| Presence of TIPS | 185 (3) | 130 (4) | 0.06 |

Continued

| Table 1 Continued | Weekday transplant | Weekend transplant | P value |
|---|---|---|---|
| | N (%) unless otherwise stated | | |
| Sepsis confirmed | 224 (4) | 175 (5) | 0.001 |
| Portal vein thrombosis | 104 (2) | 61 (2) | 0.9 |
| BMI kg/m$^2$, mean (SD) | 26.4 (5) | 26.4 (5) | 0.5 |
| **Donor characteristics** | | | |
| Donor age, mean (SD) | 46.4 (15) | 46 (16) | 0.3 |
| DCD | 596 (11) | 333 (10) | 0.7 |
| Split liver | 398 (7) | 190 (6) | 0.04 |
| Organ appearance suboptimal | 1208 (22) | 653 (20) | 0.2 |
| Cause of death | | | |
| *CVA* | 3736 (67) | 2174 (68) | |
| *Miscellaneous* | 1186 (21) | 572 (18) | 0.0002 |
| *RTA* | 410 (7) | 293 (9) | |
| *Other trauma* | 281 (5) | 164 (5) | |
| **Operative characteristics** | | | |
| Night-time procurement | 3799 (68) | 2429 (76) | <0.0001 |
| Night-time transplant | 3715 (66) | 2386 (74) | <0.0001 |
| CIT (hours), mean (SD) | 9.3 (3) | 9.4 (3) | 0.24 |
| Previous abdominal surgery | 1115 (20) | 670 (21) | 0.2 |
| **Failure** | | | |
| Overall graft failure (%) | | | |
| 30 days | 6 | 5 | 0.08 |
| 1 year | 11 | 10 | 0.16 |
| 3 years | 15 | 14 | 0.16 |
| Overall transplant failure (%) | | | |
| 30 days | 8 | 7 | 0.01 |
| 1 year | 16 | 14 | 0.09 |
| 3 years | 22 | 20 | 0.12 |
| **Cause of death at 30 days** | **229 (4)** | **120 (4)** | |
| Cardiothoracic/ myocardial ischaemia and infarction | 29 (13) | 17 (14) | |
| CVA | 11 (5) | 2 (2) | |
| Haemorrhage | 15 (7) | 7 (6) | 0.3 |
| Infection/septicaemia | 32 (14) | 15 (13) | |
| Multisystem failure | 67 (29) | 47 (39) | |
| Other | 75 (33) | 32 (27) | |
| **Cause of graft failure at 30 days** | **314 (6)** | **152 (5)** | |
| Biliary complications | 6 (2) | 4 (3) | |
| Hepatic artery thrombosis | 95 (30) | 42 (28) | |
| Non-thrombotic infarction | 14 (4) | 12 (8) | 0.6 |
| Primary non-function | 93 (30) | 50 (33) | |
| Rejection | 7 (2) | 3 (2) | |
| Other | 99 (32) | 41 (27) | |

BMI, body mass index; CIT, cold ischaemia time; CVA, cerebrovascular accident; DCD, donation after cardiac death; MELD, model of end-stage liver disease; RTA, road traffic accident; UKELD, UK model for end-stage liver disease.

To assess the fit of the models at each endpoint, we used the May and Hosmer test.[45] Schoenfeld residuals and the Grambsch and Therneau test were used to test the assumption of proportional hazards. The functional form of each continuous variable in the model was assessed for non-linearity using martingale residual plots from the null model and by fitting spline terms. The predictive ability of the models was summarised using the c-statistic.[46]

Data supplied to the UK Transplant Registry are validated on receipt to ensure completeness of follow-up. Transplant centres are contacted directly if there are validation queries, or to obtain complete data records. Patient survival is confirmed through death registration where possible. All analyses were performed using SAS V.9.4 (SAS Institute).

### Patient involvement

Patients, carers and lay people were not directly involved in the design, conduct and analysis of this study, as it is based on routinely collected data from the UK Transplant Registry. However, the study was designed to assess outcomes that are important to patients including graft failure and mortality. The relevance and timeliness of the study was endorsed by the NHS Blood and Transplant Liver Advisory Group which includes transplant clinicians, patients and lay members.

### RESULTS

Data were available for 8816 adult LT performed at all UK centres. Follow-up information on graft failure or patient death was available at 30 days for all transplants, at 1 year for 91.4% and at 3 years for 76.2% of transplants. Follow-up information is obtained from annual follow-up appointments with patients, or notification of death. In the analysis, follow-up information was censored at the last known follow-up for the patient within the follow-up period of analysis (30 days, 1 or 3 years post-transplant).

The mean recipient age (SD) at transplantation was 50 (13) years, 60% of the population were male and 88% were Caucasian. Alcohol-related liver disease was the leading indication for transplantation (20%) followed by chronic viral hepatitis (16%), acute liver failure (12%), retransplantation (10%), primary biliary cholangitis (10%), primary sclerosing cholangitis (8%), autoimmune hepatitis (8%), primary liver cancer (6%), metabolic liver disease (6%) and other diagnoses (5%). The donor population were 51% male with a mean average age (SD) of 46 (16) years. The mean CIT (SD) was 9.3 (3) hours. Eleven per cent of livers were from DCDs and the remainder from DBDs.

Of the 8816 transplants, 3203 (36.3%) were performed in the weekend period and 6101 (69.2%) at night. Overall unadjusted transplant failure estimates were 7.9% at 30 days, 15.3% at 1 year and 21.3% at 3 years and graft failure estimates were 5.7%, 10.4% and 14.6%, respectively.

### Effect of weekend transplantation

Table 1 summarises the donor and recipient characteristics by weekday and weekend transplantation which were all tested for inclusion in the model building process.

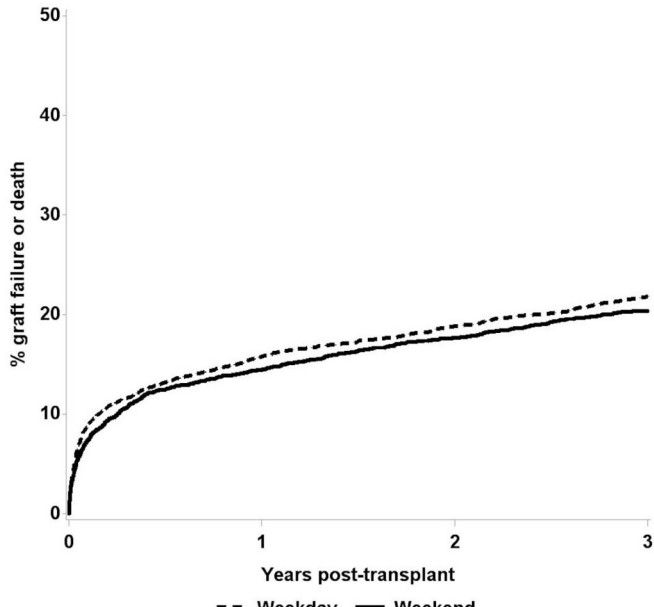

**Figure 1** Unadjusted transplant failure up to 3 years post-transplant.

Transplants at the weekend had a higher frequency of factors associated with poorer outcomes including being an inpatient, being listed for super-urgent indications and having active sepsis at the time of transplantation, however more split liver grafts were performed on weekdays. Lower mean average MELD and UKELD scores were seen on weekdays compared with weekends (19.1 vs 19.5 (p=0.05) and 56.0 vs 56.4 (p=0.02), respectively) although the difference is small and may not be clinically meaningful. Night-time procurement and transplantation were more likely at the weekend.

Similar proportions of first transplant and retransplant procedures, organs from DCDs or DBDs and livers from paediatric or adult donors were observed between weekend and weekday procedures.

The following factors were found to be non-significant when comparing weekday and weekend transplants, and in the risk-adjusted models: recipient gender, cause of recipient death, cause of graft failure, donor gender, outcome of first offer (decline/accept).

In the unadjusted analysis, graft failure was similar for weekday and weekend transplantation with 6% and 5% graft failure at 30 days, respectively, 11% and 10% at 1 year, and 15% and 14% at 3 years. Transplant failure was higher among weekday recipients at 30 days (8% vs 7%, p=0.01), but not significantly different at other time points (see figure 1).

The factors that significantly affected transplant failure at each of the three time points and were included in each model were *recipient factors*: on renal support, ventilated, confirmed sepsis at time of transplantation, primary liver disease, age, inpatient at time of transplantation, acute liver failure, previous abdominal surgery, presence of oesophageal varices, presence of TIPS, Caucasian and *graft factors*: organ appearance suboptimal, CIT, donor

**Table 2** Cox regression model for chance of transplant failure following liver transplantation at the weekend compared with a weekday

| Time from transplant | Unadjusted | | | Risk adjusted | | |
|---|---|---|---|---|---|---|
| | HR | 95% CI | P value | HR | 95% CI | P value |
| 30 days | 0.818 | 0.698 to 0.959 | 0.01 | 0.772 | 0.658 to 0.906 | 0.001 |
| 1 year | 0.906 | 0.809 to 1.014 | 0.09 | 0.864 | 0.771 to 0.968 | 0.01 |
| 3 years | 0.925 | 0.839 to 1.020 | 0.12 | 0.893 | 0.809 to 0.986 | 0.02 |

age, DCD, split liver and transplant year and night-time transplant. Each model was built separately for the different endpoints and outcomes (graft failure or transplant failure) and any significant factors were included in the risk-adjusted models. Online supplementary table A2 and 3 demonstrate the risk-adjusted HR for each variable from the Cox regression model for the chance of transplant failure or graft failure, respectively. The c-statistics for the transplant failure and graft failure models were 0.65, 0.63 and 0.60; and 0.64, 0.62 and 0.60 at 30 days, 1 year and 3 years, respectively.

The risk-adjusted HR (95% CI) of transplant failure for weekend transplant relative to weekday was 0.77 (0.66 to 0.91) within 30 days. The corresponding HRs for 1 year and 3 years were 0.86 (0.77 to 0.97), 0.89 (0.81 to 0.99), respectively (table 2). A weekend transplant had a significant effect on hazard of graft failure alone at 30 days post-transplant of 0.81 (0.67 to 0.97), and marginal at 1 and 3 years post-transplant (table 3). To ensure that the imputed data did not influence the outcomes, analysis excluding the cases with imputed data (therefore including 8037 cases) revealed a similar pattern of results (data not shown).

Differences in surgical complexity could potentially have influenced outcomes between time periods if, for example, more complex patients were selected for transplantation during the week. There is no direct measure of surgical complexity available, but factors that may reflect this were similar between weekdays and weekends including the mean number of units of blood transfused intraoperatively (5.1 weekdays vs 5.0 weekends, p=0.6), mean length of inpatient stay (21.9 days and 22.4 days, respectively (p=0.24)), presence of portal vein thrombosis (2% vs 2%, p=0.9), recipient BMI (26.4 vs 26.4 kg/m², p=0.5), presence of TIPS (3% vs 4%, p=0.06) and prior abdominal surgery (20% vs 21%, p=0.2), although the mean length of ICU stay was longer with weekend transplantation (5.4 (SD 9) vs 6.1 (SD 12) days, p=0.008).

To explore whether there were specific periods in the week that were associated with an increased risk of transplant or graft failure, we tested early and late weekdays and each day individually. There was no observed difference in graft or transplant failure between patients transplanted during the day in the early week (Monday 08:00–19:00 hours, Tuesday and Wednesday 07:00–19:00 hours) and the late week (Thursday 07:00–19:00 hours and Friday 07:00–17:00 hours), furthermore the day of the week did not significantly affect the risk of failure after risk-adjustment at any of the three time points. An interaction between weekend and night-time transplant was found to be non-significant (p=0.9 at 30 days, 0.5 at 1 year and 0.7 at 3 years). As there were large differences in the number of transplants performed per centre during the study period (range 533–2112), we tested whether there were variations in the frequency of weekend transplantation and in weekend outcomes on a per centre basis. The weekend effect was consistent across all transplant centres, and an interaction between weekend transplant and centre was non-significant (p=0.3 at 30 days, 0.2 at 1 year and 0.1 at 3 years).

We considered whether there were different rates of higher risk organ use between weekends and weekdays. We observed that use of organs previously declined by another centre at their first offering was the same for weekdays (31%) and weekends (32%) (p=0.23) and the proportion declined due to donor factors, as compared with other factors, was identical. Donor age, use of DCDs, CIT and suboptimal organ appearance were similar.

To determine when the factors leading to reduced outcomes following weekday transplantation were operating, analysis was restricted to those who did not suffer death or graft failure prior to 30 days. This resulted in no significant weekend effect on transplant failure after one and 3 years, indicating that the factors operate within the first month following transplantation and continue to affect long term failure rates.

**Table 3** Cox regression model for chance of graft failure following liver transplantation at the weekend compared with a weekday

| Time from transplant | Unadjusted | | | Risk adjusted | | |
|---|---|---|---|---|---|---|
| | HR | 95% CI | P value | HR | 95% CI | P value |
| 30 days | 0.845 | 0.700 to 1.020 | 0.08 | 0.805 | 0.665 to 0.973 | 0.02 |
| 1 year | 0.905 | 0.787 to 1.039 | 0.16 | 0.874 | 0.760 to 1.005 | 0.06 |
| 3 years | 0.918 | 0.814 to 1.036 | 0.16 | 0.892 | 0.790 to 1.007 | 0.06 |

## Effect of night-time procurement or transplantation

A similar analysis and model building strategy was undertaken for transplants assessed as day-time (operative start time 07:00–19:00 hours) compared with night-time and also for day-time compared with night-time organ procurement. A total of 2715 (31%) transplants were undertaken during the day-time and 6101 (69%) at night. Night-time procurement occurred in 6228 (71%) of transplants. Day-time compared with night-time transplant was associated with lower graft failure at 30 days (5% vs 6%, p=0.02), 1 year (9% vs 11%, p=0.007) and 3 years (13% vs 15%, p=0.002) and transplant failure at 30 days (7% vs 8%, p=0.008) and 1 year (13% vs 16%, p=0.0004) and 3 years (19% vs 22%, p=0.001) in the unadjusted analysis. Day-time compared with night-time procurement was associated with lower graft failure at 30 days (5% vs 6%, p=0.04), 1 year (9% vs 11%, p=0.005) and 3 years (13% vs 15%, p=0.002) and transplant failure at 30 days (7% vs 8%, p=0.03), 1 year (13% vs 16%, p=0.0007) and 3 years (19% vs 22%, p=0.003) in the unadjusted analysis. In the Cox proportional hazard models, using the same variables as for the weekend versus weekday model neither night-time procurement nor night-time transplantation had a significant effect on transplant or graft failure at any time point (online supplementary table A4 and 5).

## DISCUSSION

We have demonstrated no increased risk of graft or transplant failure with weekend LT, night-time LT or night-time graft procurement. These findings suggest that UK liver transplant outcomes do not have an increased risk of adverse outcomes associated with 'out-of-hours' operating. Interestingly, there was a possible reduction in the hazard of early graft failure and long-term transplant failure associated with weekend transplantation. The loss of this association when considering only survivors at 30 days suggested that any responsible factors were acting in the peritransplant and early post-transplant period. The effect was present in the unadjusted transplant failure data at 30 days, in the adjusted outcomes at all time points measured and was seen across all centres.

There is a wide range of putative confounding factors that may have influenced the risk of graft and transplant failure during in-hours and out-of-hours transplantation. We attempted to control for those that were measured in the database including donor, operative and recipient characteristics (as listed in table 1). Unmeasured or non-quantifiable confounders such as risk aversion in operator practice, clinical team structure or seniority, pressure on general hospital resources, donor graft quality and patient fitness may still have been operating, potentially confounding our observations. As explained below, we have attempted to identify proxies for these where possible, but due to the complex, multifactorial influences on liver transplant outcomes we could not exclude the presence of confounding effects.

To explore whether the finding of possible improved weekend outcomes could be explained we tested several hypotheses. Patients transplanted at weekends were not lower risk, in fact they were more likely to have adverse prognostic markers and there were no differences in the causes of liver disease between the two groups. There were no systematic differences in organ utilisation; markers of adverse graft features were similar including average donor ages and the proportion of DCD, suboptimal and previously declined organs that were transplanted. No differences in markers of surgical complexity were noted including portal vein thrombosis, oesophageal varices, retransplantation, prior abdominal surgery, BMI and volume of blood transfused intraoperatively. There was no evidence for systematic delay of operative start times following organ retrieval as the CIT was similar. If this were occurring to ensure adequate practitioner rest at night or prioritise elective commitments, it would theoretically have been seen more in weekday transplantation.

A lead-in effect, where patients transplanted in the week are stepped down from the ICU to general wards over the weekend, would disproportionally affect transplants occurring early in the week, however we found no such association. Weekend LT was associated with a longer average ICU stay (6.1 vs 5.4 days) but whether this would improve outcomes, was a marker of a more unwell patient population or was of any clinical importance is unclear. Therefore, we are unable to explain this association and it may relate to other unmeasured factors in patient or graft selection or care.

We speculate that the senior clinician-led and delivered service may obviate any inherent weekend risks. Although as previously noted, not all clinical scenarios are associated with 'out-of-hours' risks. However, this service structure would not explain the difference we observed between weekend and weekday outcomes. It is conceivable that there may be a protective effect of weekend staffing patterns or the hospital environment on LT outcomes, for example, the absence of routine elective work and competing clinical activity may potentially free clinicians and resources for a more prompt and responsive service.

We have demonstrated no increased hazard of graft or transplant failure for liver transplants performed out-of-hours. Whether this was due to weekends being inherently more risky (due to, for example, operator fatigue, reduced staffing and ancillary support services) but that the senior-led service and lack of competing clinical activity prevented this from resulting in poor patient outcomes, or conversely that in-hours and out-of-hours transplantation carried a similar baseline risk, cannot be unpicked in this study. If the inverse weekend effect we observed is true, we believe the modifying factor likely lies in differences in staffing and competing clinical activity as outlined above.

The limitations of this work include that, as with all retrospective registry based studies, neither causality nor aetiological factors can be identified. Furthermore, registry data are dependent on the quality of data imputing,

curation and assignation of variables. However, this is a well curated, highly complete dataset that has previously been used for multiple studies. Secondly, the definition of day and night with regard to transplantation surgery was artificial as combined organ retrieval, transfer and implantation straddles both day and night periods, however our methodology has been utilised in a previous similar study of LT.[34] Due to the large size of the database, some observed differences will reach statistical significance despite being clinically insignificant (for example, the statistically significant but very small difference observed in UKELD score between weekday and weekend transplantation groups) and as such differences should be interpreted with caution. Likewise, due to the modest difference in graft and transplant survival observed between these two groups, we suggest that there was, at the least, no evidence for worse outcomes at the weekend.

This study is an interesting comparator for published disease specific and unselected admissions studies of out-of-hours outcomes.[3–17 20–24 27 28] Our cohort of patients had a standardised risk profile throughout the week unlike unselected admission studies, as to qualify for LT they had to meet minimum clinical thresholds, assessed by objective validated clinical scores, yet had to be well enough for the procedure to be performed, thus obviating the weekday compared to weekend selection bias inherent in hospital admission studies. Furthermore, the availability of an organ, rather than a patient's clinical status, determined the timing of the admission and clinical intervention, in contrast to unselected admission studies. Finally, our study benefited from a detailed, well-curated database of individual patient clinical parameters that enabled the construction of accurate risk-adjustment models to correct for variation in risk on a per patient basis.

The non-inferiority of out-of-hours LT, as seen in other LT and renal transplant studies,[34–38] is reassuring and illustrates that the current model of liver transplant provision in the UK provides acceptable outcomes at traditionally perceived periods of clinical risk. However, the potential for improved weekend clinical outcomes differs to that seen in other studies on LT.[34 35] A potential beneficial weekend effect is interesting as, if working patterns or hospital resources are responsible, this represents a model for improved patient care. Direct comparison of surgeons', ICU and physicians' workload, clinical commitments, and working patterns between weekdays and weekends and comparisons with other, senior-clinician-led services will be of interest.

In summary, we have demonstrated that 'out-of-hours' LT outcomes are not worse than for 'in-hours' procedures, and that potentially weekend LT may be associated with reduced adverse outcomes. The weekend LT care structure in the UK may represent a model for the design of other critical out-of-hours services. Furthermore, these findings illustrate the complexity of observed weekend effects which are likely to be dependent on patient selection.

**Contributors** NH designed the study concept and strategy, undertook data analysis and drafted and revised the manuscript. KM proposed statistical methods, undertook data analysis and drafted the manuscript. DC proposed statistical methods, reviewed data analysis and drafted the manuscript. EA proposed statistical methods, undertook data analysis and drafted the manuscript. DT designed the study concept and strategy, undertook data analysis and drafted and revised the manuscript.

**Funding** NH is supported by a Wellcome Trust PhD Studentship.

**Competing interests** None declared.

**Patient consent for publication** Not required.

**Ethics approval** Ethical approval was not required as this study relied solely on retrospective analysis of pseudanonymised patient data collected for the purposes of clinical care and programme outcome evaluation.

**Provenance and peer review** Not commissioned; externally peer reviewed.

**Data sharing statement** Data and statistical code may be made available on request by enquiry via the corresponding author, following approval by NHS Blood and Transplant.

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
