## [Reviewer comments · BMJ Open]

ARTICLE DETAILS

TITLE (PROVISIONAL)	Is liver transplantation 'out-of-hours' non-inferior to 'in-hours' transplantation? A retrospective analysis of the UK transplant registry.
AUTHORS	Halliday, Neil; Martin, Kate; Collett, David; Allen, Elisa; Thorburn, Douglas

VERSION 1 – REVIEW

REVIEWER	John Peipert Northwestern University, USA
REVIEW RETURNED	20-Jul-2018

GENERAL COMMENTS	Given the conflicting evidence on off-hours solid organ transplants, this is a well-needed study. Since it uses country-wide data over a long period of time, the design is strong. The results largely demonstrate that there are no differences between off-hours and normal hours LT in the UK. Though some results are statistically significant, actually favoring off-hours LT, the differences are modest and the large sample size may foster statistically significant differences for even small, clinically non-significant differences. I suspect that is the case for this paper. I am enthusiastic about this study, but there are a few methodological questions to address. Specific comments are attached.
---

REVIEWER	Neri Alejandro Alvarez Villalobos Universidad Autonoma de Nuevo Leon, Subdireccion de Investigación
REVIEW RETURNED	21-Jul-2018

GENERAL COMMENTS	Dr. Neil Haliday et al. report a retrospective analysis of a Liver Transplantations database from all the United Kingdom centers in which they aimed to assess the impact of night-time and weekend transplantations on major outcomes such as recipient death and graft failure in comparison to day-time and weekday transplantations. The study is prone to measurement bias since the whole study data originates from a database, however since less than 5% of the data is missing, and the database is well organized, the risk of bias would be low. The study question is interesting and concise; the manuscript is well written and easy to understand. Nevertheless, there are a few aspects that need to be clarified: 1. In the methods section, the authors mention that data from all adults 17 or older was retrieved. Nonetheless, 17-year-olds are not considered as adults according to the WHO.
--

	2. There are some specific statistical issues to clarify, specifically why the Student T-test was used. It looks like sometimes apply a non-parametric test (example: comparing "ICU stay"). I would suggest to clarify variables distribution and the use of T-test, as well as reporting with median and interquartile interval or range when it applies. 3. Could the authors explain the potential confounding variables and effect modifiers of the risk of graft failure and transplantation failure when comparing night-time vs. day-time and weekday vs. weekend transplantation?
--	--

REVIEWER	Dag Olav Dahle Department of Transplantation, Oslo University Hospital, Rikshospitalet, Oslo, Norway
REVIEW RETURNED	06-Aug-2018

GENERAL COMMENTS	Thank you for the opportunity to review this manuscript, with an emphasis on the statistical methodology. The aim is stated in the title; "Is liver transplantation 'out-of-hours' non-inferior to 'in-hours' transplantation? A retrospective analysis of the UK transplant registry"; I think this is an important research question. The study is large, thorough and has novel and important findings. I hope the comments below are helpful to strengthen the manuscript. As this review is open I have elaborated somewhat on my concerns. 1. Regarding the aim: The aim is clearly stated in the abstract, where it is in line with the analyses undertaken, but the aim is also stated in the title (as mentioned) and at the end of introduction (to assess "the impact of night-time and weekend LT [liver transplantation] upon recipient death and graft failure following single organ LT across all UK centres). Here, the aim statement is slightly imprecise, since the study did not use 'in-hours' (ie. week-day daytime) LT as comparator for any analysis. Rather, two analyses are shown, comparing night with day, and week-end with week-day. Consider, perhaps as supplemental, one analysis with four exposure groups (week-day daytime as comparator/reference, week-day night, week-end day, week-end night), this would also show interaction. Also, regarding the aims statement in the introduction "recipient death" was not assessed as a separate outcome. 2. Regarding the exposure: timing of LT is a proxy for the real exposure(s), which would be staff or system factors varying with time-of-day and time-of-week, as you describe in the introduction. From a clinical perspective, I think a very relevant risk factor/exposure to document would be human error due to tiredness and fatigue at night/long week-ends and/or due to less experienced clinicians on call. Thus, ideally, the study could use day/night shift status of the surgeon as exposure, and experience of the surgeon as another exposure. Using timing of LT as proxy, I think it is imperative that the definition of night-time covers operations undertaken by the night shifts. The definition of night-time was a bit hard to understand (P 8 lines 1-19). First, the "liver perfusion start time" is ambiguous to a non-surgeon, to make sense this must be start of cold perfusion and start of cold-ischemia time. Second, the "operation time" seem to be defined by re-perfusion (end of CIT) occurring between 7pm-7am. However, LT with re-perfusions at 8am are perhaps also carried out by night-shift teams, (at least the procurement procedure), and the study
---

	should present (or at least mention) sensitivity analyses using other relevant cutoffs segregating night and day shifts. 3. Regarding the endpoints: Graft failure and transplant failure are semantically synonymous and not defined in the abstract. Both are combined endpoints, as defined p 8 line 24-33. Transplant failure is defined as the earlier of graft failure before death, graft failure and death, or death with a functioning graft. Graft failure is graft failure before death or graft failure and death. Thus, the handling of death with a functioning graft is what separates these two endpoints. Perhaps “graft loss” (GL) and “graft loss or death with a functioning graft” (GLD) would be easier to understand and remember for readers? Consider clarifying how re-transplantation fits into these definitions. 4. Regarding missing data on predictors: There is little missing data in the predictors, <5%, as detailed in table A1 p28, though you did not state how many patients were complete on all covariates. You wisely use a multiple imputation procedure in the main analysis to avoid deleting patients with missing predictor values. However, according to the methods section p9 line 16 it seems you generated one full/final data set to run the analyses in. Multiple imputation usually involves generating multiple datasets, run the analysis in all of them and averaging/aggregating the result at the end (taking into account measurement error between the imputed datasets). You may also consider including the outcome variables (both status and time variable for survival data) when making the imputed datasets. Nevertheless, at this low level of missing data, a full-case analysis usually reveals very similar results as a correctly undertaken multiple imputation analysis, as you indicate on page 13 line 14. 5. Regarding missing data on outcome. You state there were missing outcome data in the methods section (p 9 line 27) and in the results section you state there were 91,4% follow-up data at one year and 76,2% at three years. How many were in fact lost to follow-up and how did you handle them in the analysis? Usually they are censored on last follow-up. 6. Regarding the models.  -The number of events determines the power of survival analysis, not the number of patients. Thus, for the reader to be confident in the analysis, the number of events should be stated. To avoid model optimism, about 10 events are needed for each predictor evaluated for inclusion in the model (ie. not only per predictor in the final analysis.) - Variable selection procedures using the outcome (ie using the Cox model) tend to retain some spuriously strong predictors, which also inflate the C-statistic. If you are to screen for confounder variables to include in a final model, consider screening against the exposure instead (baseline table; since confounders are related to the exposure as well as the outcome). If the number of events is very large, perhaps no variable selection procedure is necessary. -For graft failure, patient death with a functioning graft is a competing event. Estimating the probability for graft failure is thus best undertaken with a competing risk model (Noordzij et al NDT 2013). For the main aetiological analysis of HRs, however, standard Cox is preferred. -You did not state in the methods how you analyzed time-dependency of the predictor, ie. how you got the different HRs for the different time periods. For the later time points, it seems you included the initial follow-up also. Consider for later time-points excluding initial follow-up for instance using a heaviside function
--	--

	(see for instance Kleinbaums "Survival analysis", Ch 6). Since the survival curves are more or less parallel after the initial follow-up I assume the proportional hazards assumption does not hold (the HRs should be more or less 1 after initial follow-up.) 7. Regarding the results. -Consider presenting day/night analyses in a similar way and similar tables and figure as week-end/week-day, for instance in supplemental material. Results mentioned in tables need not be repeated in the text. -You unexpectedly find less transplant failure (ie graft failure or death) at 30 days for week-end LT (p12 line 13), even though these patients' were sicker (p11 line 29), and this held true in both unadjusted and adjusted models. At the same time, you find similar unadjusted (death-censored) graft failure rates at weekday and weekend (p12 line 13). These are very interesting findings, and I appreciated you attempt to find explanations. Further discussion on this issue is beyond the scope of a statistical review, but you may consider including a separate analysis of patient death. 8. Minor comments: -The many analyses incorporated into this manuscript make it a bit hard to read. Consider focusing the text on the major aetiological analysis (HRs), event probabilities could be presented in tables beneath the survival curves. -Table A2 showing the full results for all risk factors is not explained, - HRs in bold seem to stem from the multivariable model, and the non-highlighted are then probably univariate? Consider excluding this table.
--	---

REVIEWER	Wenceslao Peñate University of La Laguna, Tenerife, Spain
REVIEW RETURNED	19-Aug-2018

GENERAL COMMENTS	The manuscript deals with "out-of-hours" clinical practice (weekend vs. weekdays; night-time vs. day-time), on liver transplantation (success vs. failure). Authors used a very relevant archive sample. Authors used a very relevant archive sample. Statistical procedures are suitable and can answer research questions. I only have some questions about data presentation: Authors state there is not increased risk of failure with weekend or night-time surgical intervention. The unique exception is in the 30-days interval, where a higher failure is observed in weekday's interventions. But if we observe table 2, adjusted models are also significant for one year (p=0.01) and three years (p=0.02). There is a debate about p values in clinical data (there are proposals for p values lower than .005 for statistical significance; i.e. http://dx.doi.org/10.1001/jama.2016.2152; http://dx.doi.org/10.1001/jama.2018.1536). If authors agree with that proposal, I think it will be useful to declare it in data analysis subsection (and applied it to the rest of analyses). Also, the authors find several differences between patients with success / failure in liver transplantation. I think manuscript can provide relevant information about factors associated to failure / success liver transplantation, if auxiliary other data analyzes are performed. I. e., Logistic regression on significant variables (better, mixed linear analysis, because of several data are nested). In any case, the manuscript is sufficiently relevant as it is.
--

VERSION 1 – AUTHOR RESPONSE

Reviewers' Comments to Author:

Reviewer: 1

Given the conflicting evidence on off-hours solid organ transplants, this is a well-needed study. Since it uses country-wide data over a long period of time, the design is strong. The results largely demonstrate that there are no differences between off-hours and normal hours LT in the UK. Though some results are statistically significant, actually favoring off-hours LT, the differences are modest and the large sample size may foster statistically significant differences for even small, clinically non-significant differences. I suspect that is the case for this paper. I am enthusiastic about this study, but there are a few methodological questions to address.

- The introduction reviews a good deal of literature that could be condensed and made more concise. Focus should be placed on the solid organ transplant studies and reduce the general literature to a few sentences.

Many thanks for this comment. Our intention was to cover the literature surrounding out-of-hours clinical practice more broadly as this study sits between several fields of research and clinical practice: patient safety and quality, solid organ transplantation, surgery, intensive care and hepatology. Due to the range of clinical arenas that have been studied in the literature, there is often extrapolation of findings from one area to another. We recognise that the introduction could be more focused, as suggested, and have revised the manuscript accordingly.

- The statistical methods are strong. However, the approach to modeling could be clarified a bit. It is clear that there were unadjusted and adjusted Cox models, some of the results language seems to indicate that you may have examined whether some covariates modified the association between off- vs. normal hours LT and outcomes. E.g., p. 12, lines 3-10 and the abstract conclusions stating there is a possible beneficial effect dependent (implying effect modification) upon peri-transplant factors. Was this tested with interaction terms between time period cohort (off vs. normal hours) x covariates? If so, I did not see in the methods, but this type of analysis may enrich the results if tested.

The statement that any potential superior outcomes observed at the weekend were dependent upon peri-transplant factors was derived from the observation that the difference between outcomes for weekend and weekday LT disappeared if only patients who survived with a functioning graft to 30 days post transplant were considered. This suggests that, if the improved outcomes are real, the deaths and graft loss that occur in the weekday group must be occurring within 30 days of transplantation and therefore be driven by clinical events occurring within this time period. We have adapted the statements in the abstract, the results (final paragraph of Effect of Weekend Transplantation section) and discussion to make this clearer.

- As noted in my overall comments, due to the rather large sample size, even small, clinically nonsignificant differences could be statistically significant. The Discussion should mention this possibility unless a power analysis is conducted to determine that statistical power is not extremely high. Similarly, given the modesty of differences in outcomes between off- and normal hours LT, consider scaling back suggestion in the abstract that there is a beneficial effect for weekend transplant unless this can be further explained in the Discussion.

Many thanks for this comment; we had considered size of clinical effect in addition to the statistical significance and tried to avoid overstating small, but statistically significant differences. It is not appropriate to carry out a power analysis once the study data are available, and as we used all data

from UK patients in a chosen time period, and had little or no a priori knowledge about the difference that we might detect, we did not carry out a power calculation beforehand. The fact that the improved weekend outcomes were observed consistently across all units, were present in the unadjusted and adjusted models and were dependent upon events occurring at a specific and clinically relevant timeframe (i.e. immediate to the transplantation) we do not feel that this observation can be overlooked. However, we accept that likewise it should not be overstated as there are potential unmeasured and uncorrected confounders and a modest clinical effect. We have softened the language used in the abstract and included additional caveats in the discussion to that effect.

- The overall time period from which transplant records were drawn was somewhat long: 2000- 2015. This is mostly a strength, but changes in practice of that time period could be related the results. Did the authors consider examining whether year of transplant was associated use of off-hours LT, and if outcomes of off hours LT varied over this time period (or at least varied differently than on-hours outcomes).

In our exploration of data, we did consider both whether the year of transplant affected the proportion of weekend transplants and the occurrence of adverse outcomes. There was a general trend to a greater proportion of weekday transplants in more recent years (table 1). Year of transplant did significantly affect the risk of graft failure or transplant failure (supplementary figures A2 and A3) with a reducing hazard of these outcomes in more recent years.

We observed evidence of non-proportionality of the hazard function for the year of transplant variable and accounted for that by breaking down the year of transplant variable for individual years (as shown in Table A2 and A3). Our multivariate model adjusted for the stratified year of transplantation variable, as stated in the results section (Effect of Weekend Transplantation, paragraph 5). Therefore the impact of changing clinical practice over time has been accounted for and would not bias the outcomes of the study.

- Please clarify the stepwise selection procedure and criteria or rationale for entering and removing variables.

A stepwise forward selection procedure was used, and at each stage in the process we 'looked back' to see if any variables included earlier were no longer significant. This this was done 'manually', rather than using a computer-based algorithm and model choice was guided by a judicious combination of significance levels and clinical considerations. We have included this information in the methods section of the manuscript.

- The results indicate that follow-up information was missing in a non-trivial number of patients. Were these censored in the time to event analyses? If not, why not? In general, more details about censoring choices for the time to event analyses should be given.

In the analysis, follow up information was censored at the last known follow up for the patient within the follow up period of analysis (30 days, 1 or 3 years post transplant). We have included a statement to this effect in the results section of the manuscript so that this point is clear to the reader.

- Small point: what was the contribution of the t-tests and chi-square tests to compare outcomes over and above the time to event analyses?

t-tests and chi-square tests were used to determine if there were differences between a range of continuous and categorical variables between weekend or weekday transplantation. This enabled us to understand the distribution and significance of potential factors between the two periods compared. We feel this allows the reader to determine the magnitude and significance of differences in clinical

features that may be influencing patient outcomes between the operative periods (as shown in table 1). They do not add additional information beyond the final time to event analyses but we included them as we feel this makes our findings more open to the reader and allow them to assess whether their population of patients and distribution of risk factors would be comparable to our study cohort.

- Very small point: There are a few terms used in the Abstract that could be improved. 1) It is more appropriate to say that the outcomes were “associated” instead of “correlated” with weekend vs. weekday, day vs night time LT; 2) Does “altered” hazard ratio refer to “adjusted” hazard ratio? We agree that the terms used here were either not optimal, or ambiguous and they have been changed in the text of the abstract.

Reviewer: 2

Dr. Neil Haliday et al. report a retrospective analysis of a Liver Transplantations database from all the United Kingdom centers in which they aimed to assess the impact of night-time and weekend transplantations on major outcomes such as recipient death and graft failure in comparison to day-time and weekday transplantations. The study is prone to measurement bias since the whole study data originates from a database, however since less than 5% of the data is missing, and the database is well organized, the risk of bias would be low.

The study question is interesting and concise; the manuscript is well written and easy to understand. Nevertheless, there are a few aspects that need to be clarified:

1. In the methods section, the authors mention that data from all adults 17 or older was retrieved. Nonetheless, 17-year-olds are not considered as adults according to the WHO.

Many thanks for this comment. Whilst we accept that the standard WHO definition of an adult is from the age of 18, in practice in the UK patients of the age of 17 are often treated within adult clinical environments and 17 years of age defines the point at which patients can access the adult liver transplant program in the UK. For the purposes of UK liver transplant service performance and outcomes monitoring, 17 year olds are considered adult. Hence we felt it would be inappropriate to exclude this group of patients who share and access the same services and the wider cohort.

Recipient age was included in our analyses and was not found to be associated with transplant failure so we do not believe that the inclusion of 17 year olds will have influenced the outcome of the study.

2. There are some specific statistical issues to clarify, specifically why the Student T-test was used. It looks like sometimes apply a non-parametric test (example: comparing “ICU stay”). I would suggest to clarify variables distribution and the use of T-test, as well as reporting with median and interquartile interval or range when it applies.

Many thanks for considering the selection of appropriate statistical tests in this analysis. In addition to our comments here, please see our response to point 7 raised by reviewer 1. However, to directly answer the question here, t-tests were always used to compare continuous variables, including duration of ICU stay (reference is made to this in the results section of the manuscript). There was no evidence of a severe lack of symmetry in any of these variables, so that non-parametric techniques were not needed. We have therefore provided means and sd values in Table 1.

3. Could the authors explain the potential confounding variables and effect modifiers of the risk of graft failure and transplantation failure when comparing night-time vs. day-time and weekday vs. weekend transplantation?

Thank you for this observation. We spent much of the discussion exploring potential confounders and effect modifiers including recipient fitness and clinical characteristics, donor organ risk, surgical

complexity (as a marker of both potential risk aversion behaviour by clinical teams and of clinical complexity) and point of step down from ICU to general wards. We have added additional text to the discussion to explain that other unmeasured or unquantifiable confounders may exist. We have also added additional text that explores the difficulty in measuring whether out-of-hours transplantation is higher risk due to e.g. fatigue and staffing (as suggested for weekend effects observed in other settings) but that this is protected against by a senior staffing / service structure. Or conversely, that the underlying risk for LT is similar in all periods. Whilst we cannot ascribe causality for a primary effect modifier for protecting against out-of-hours risk, and potentially out-of-hours benefit, we propose it may relate to staffing and competing clinical activity.

Reviewer: 3

Thank you for the opportunity to review this manuscript, with an emphasis on the statistical methodology.

The aim is stated in the title; "Is liver transplantation 'out-of-hours' non-inferior to 'in-hours' transplantation? A retrospective analysis of the UK transplant registry"; I think this is an important research question. The study is large, thorough and has novel and important findings. I hope the comments below are helpful to strengthen the manuscript. As this review is open I have elaborated somewhat on my concerns.

1. Regarding the aim: The aim is clearly stated in the abstract, where it is in line with the analyses undertaken, but the aim is also stated in the title (as mentioned) and at the end of introduction (to assess "the impact of night-time and weekend LT [liver transplantation] upon recipient death and graft failure following single organ LT across all UK centres). Here, the aim statement is slightly imprecise, since the study did not use 'in-hours' (ie. week-day daytime) LT as comparator for any analysis. Rather, two analyses are shown, comparing night with day, and week-end with week-day. Consider, perhaps as supplemental, one analysis with four exposure groups (week-day daytime as comparator/reference, week-day night, week-end day, week-end night), this would also show interaction. Also, regarding the aims statement in the introduction "recipient death" was not assessed as a separate outcome.

Many thanks for this comment; we have adapted the aim as stated in the final paragraph of the introduction to make it more precise to the analysis undertaken.

During the statistical analysis we considered and undertook the approach of analysing four time periods (week day, week night, weekend day and weekend night). We observed no significant differences between the groups for graft survival and small but statistically significant differences in short term transplant survival that did not persist to three years (see table below). The observations are comparable to those for the analyses as presented in the manuscript. Therefore, for clarity we restricted the analysis presented and the construction of multivariate models to weekday vs weekend and day vs night.

Week-day Week-night Weekend-day Weekend-night p-value

Overall graft survival (%)

30 day 95 94 95 95 0.14

One year 90 89 91 90 0.15

Three years 86 84 87 86 0.11

Overall transplant survival (%)

30 day 92 91 93 93 0.02

One year 85 83 87 85 0.03

Three years 78 77 81 79 0.11

2. Regarding the exposure: timing of LT is a proxy for the real exposure(s), which would be staff or system factors varying with time-of-day and time-of-week, as you describe in the introduction. From a clinical perspective, I think a very relevant risk factor/exposure to document would be human error due to tiredness and fatigue at night/long week-ends and/or due to less experienced clinicians on call. Thus, ideally, the study could use day/night shift status of the surgeon as exposure, and experience of the surgeon as another exposure. Using timing of LT as proxy, I think it is imperative that the definition of night-time covers operations undertaken by the night shifts. The definition of night-time was a bit hard to understand (P 8 lines 1-19). First, the “liver perfusion start time” is ambiguous to a non-surgeon, to make sense this must be start of cold perfusion and start of cold-ischemia time. Second, the “operation time” seem to be defined by re-perfusion (end of CIT) occurring between 7pm-7am. However, LT with re-perfusions at 8am are perhaps also carried out by night-shift teams, (at least the procurement procedure), and the study should present (or at least mention) sensitivity analyses using other relevant cutoffs segregating night and day shifts.

Many thanks for your comments; we had many similar discussions during the development of this project as these definitions are highly challenging, especially in a retrospective setting. This project was developed from the UK transplant registry database, which contains a wealth of clinical data related to transplantation, as included in the study. Unfortunately, the database does not include operating surgeon details, service and shift roster structure of the units or other factors related to wider hospital resources available in off-hours periods, which, as you state, are likely to be the risk factors at play in off-hours activity. Furthermore these shift patterns are likely to vary amongst institutions and within institutions over time so could not be assessed robustly in a retrospective analysis. Whilst we fully agree that data regarding clinical error rates, operating surgeon and lead anaesthetist seniority/experience and shift patterns would be a very informative measure of some of the potential risk exposures, collecting this data was well beyond the scope of this study. Additionally attempting to collect these data retrospectively over the last 2 decades would be highly prone to bias. We are actively pursuing further projects which aim to understand how work-loads within units influence patient outcomes. Prospective assessment of work patterns, clinician seniority, operator fatigue, clinical error rates should be undertaken.

Whilst again we accept that the ‘night period’ operations would ideally be confined to those working on the night shift, in practice this is not possible. All UK liver transplant surgical teams, to our knowledge, have always had surgical teams working ‘on call’ periods lasting a minimum of 24hrs. These surgical teams may operate day and night when need arises, rather than following a day and night shift pattern as is seen in higher volume clinical activity (such as emergency medicine and acute admissions). As such, the notion of day vs night shifts is not applicable in this clinical setting. Furthermore, each unit has differing surgical team structures and rostering patterns that will have evolved over time, and again measuring this would be beyond the scope of the study. Lastly, we felt this was an unrealistic and arbitrary division, as stated in the manuscript, as the organ retrieval process, transport and pre-operative care, transplant operation itself and immediate ICU care will frequently span more than 12 hours. The day night split was an attempt to see if there were stark differences in the outcomes, which we did not detect.

Operative start time is again a challenging definition: in the registry there is no systematically recorded operative start time. In its absence, the most robust measure recorded in the database was a combination of cold ischaemia time and liver perfusion time. The donor liver perfusion time is the earliest measure of when the organ was retrieved from the donor and perfused with tissue preservation fluid. The cold ischaemia time is the duration from this point to the reperfusion of the graft by the recipient’s circulation. This measures a time well within the transplantation operation itself. However, this was the only consistent measure that is available in the collected data. Additionally, as discussed above, transplant units do not operate on a simple day/night shift division, so although a

reperfusion (therefore total operation as measured in our dataset) may occur at early in the day time and the operation had therefore begun during the night, the border between day and night working is indistinct so should not influence our measured outcomes. We have updated the definition of operative start time to make it clearer to the reader. Furthermore, we used this definition as it is comparable with the only other large multicentre study in this field (Orman et al, Liver Transpl 2012, 18;558-65).

Whilst we do not feel that we can improve upon these challenging definitions, we thank the reviewer for raising these points and hope that our explanations are satisfactory.

3. Regarding the endpoints: Graft failure and transplant failure are semantically synonymous and not defined in the abstract. Both are combined endpoints, as defined p 8 line 24-33. Transplant failure is defined as the earlier of graft failure before death, graft failure and death, or death with a functioning graft. Graft failure is graft failure before death or graft failure and death. Thus, the handling of death with a functioning graft is what separates these two endpoints. Perhaps “graft loss” (GL) and “graft loss or death with a functioning graft” (GLD) would be easier to understand and remember for readers? Consider clarifying how re-transplantation fits into these definitions.

Retransplantation was included in the study as a type of Primary Liver Disease (PLD); PLD is a covariate in the risk-adjusted model. We felt this was an important group of patients, who are recognised to be more complicated from a surgical perspective and have higher associated risks. Additionally they represent approximately 10% of the liver transplants undertaken during the study period, hence we felt they should be included. A patient who undergoes retransplantation will have had a graft failure event associated with their initial transplant by definition. We have included this in the definitions to make this clearer to the reader.

We utilised the terms graft failure and transplant failure as they are well defined terms within the field. Graft failure, referring to the failure of the organ transplanted (with or without death) is a core term in the transplant literature. The term transplant failure, which is where the intended outcome of transplantation fails (i.e. intended survival with a functioning organ), through loss of the graft and/or patient death is an important metric and the term is used through the transplant literature e.g.:

Goldstein BA, Thomas L, Zaroff JG, Nguyen J, Menza R, Khush KK. Assessment of Heart Transplant Waitlist Time and Pre- and Post-transplant Failure: A Mixed Methods Approach. *Epidemiology*. 2016 Jul;27(4):469-76

Rose C, Sun Y, Ferre E, Gill J, Landsberg D, Gill J. An Examination of the Application of the Kidney Donor Risk Index in British Columbia. *Can J Kidney Health Dis*. 2018 Mar 19;5:2054358118761052. doi: 10.1177/2054358118761052

Hence these terms were employed as outcomes within the study. We have defined our main outcome measures in the abstract to make this clearer for the reader.

4. Regarding missing data on predictors: There is little missing data in the predictors, <5%, as detailed in table A1 p28, though you did not state how many patients were complete on all covariates. You wisely use a multiple imputation procedure in the main analysis to avoid deleting patients with missing predictor values. However, according to the methods section p9 line 16 it seems you generated one full/final data set to run the analyses in. Multiple imputation usually involves generating multiple datasets, run the analysis in all of them and averaging/aggregating the result at the end (taking into account measurement error between the imputed datasets). You may also consider including the outcome variables (both status and time variable for survival data) when making the imputed datasets. Nevertheless, at this low level of missing data, a full-case analysis usually reveals

very similar results as a correctly undertaken multiple imputation analysis, as you indicate on page 13 line 14.

We agree that there are a number of possibilities when amalgamating imputed data sets. Our approach of using the median of continuous variables and the modal value of the categorical variables, over the imputed data sets, to give a single complete data set, was adopted for its straightforwardness.

5. Regarding missing data on outcome. You state there were missing outcome data in the methods section (p 9 line 27) and in the results section you state there were 91,4% follow-up data at one year and 76,2% at three years. How many were in fact lost to follow-up and how did you handle them in the analysis? Usually they are censored on last follow-up.

All patients were censored at last follow up (please see response to reviewer 1, point 6). The falling follow up rate over time relates to the greater chance of a patient's follow up ceasing as time progresses.

6. Regarding the models.

-The number of events determines the power of survival analysis, not the number of patients. Thus, for the reader to be confident in the analysis, the number of events should be stated.

To avoid model optimism, about 10 events are needed for each predictor evaluated for inclusion in the model (ie. not only per predictor in the final analysis.?)

The numbers of outcome events (graft failure and transplant failure) are available in table 1 under the heading "Failure" where the percentage of affected patients are listed and population sizes listed at the top of the columns. Therefore the lowest number of events would be 160 occurrences of graft failure in the weekend transplanted cohort.

- Variable selection procedures using the outcome (ie using the Cox model) tend to retain some spuriously strong predictors, which also inflate the C-statistic. If you are to screen for confounder variables to include in a final model, consider screening against the exposure instead (baseline table; since confounders are related to the exposure as well as the outcome). If the number of events is very large, perhaps no variable selection procedure is necessary.

Agreed. Please see response to Reviewer 1 on variable selection.

-For graft failure, patient death with a functioning graft is a competing event. Estimating the probability for graft failure is thus best undertaken with a competing risk model (Noordzij et al NDT 2013). For the main aetiological analysis of HRs, however, standard Cox is preferred.

We agree that a competing risks approach is a possibility, but this is unlikely to lead to inferences that are very different from the approach we used.

-You did not state in the methods how you analyzed time-dependency of the predictor, ie. how you got the different HRs for the different time periods. For the later time points, it seems you included the initial follow-up also. Consider for later time-points excluding initial follow-up for instance using a heaviside function (see for instance Kleinbaums "Survival analysis", Ch 6). Since the survival curves are more or less parallel after the initial follow-up I assume the proportional hazards assumption does not hold (the HRs should be more or less 1 after initial follow-up.)

Many thanks for raising this point, we agree that we could have been clearer in the text. Hazard ratios for different time periods were found by including a period factor in the model and we have amended the text of the methods section to say this. We accept that survival conditional on, say, survival to one month could lead to different inferences about the weekend effect. We have not included this analysis, as our interest is in the effect of out of hours surgery on the subsequent time from transplant to an event. Please also see our discussion in response to reviewer 1, point 4 regarding non-proportionality of the hazard function for the year of transplant variable and point 2 regarding conditioning of analysis for patients surviving beyond thirty days post transplantation.

7. Regarding the results.

-Consider presenting day/night analyses in a similar way and similar tables and figure as week-end/week-day, for instance in supplemental material. Results mentioned in tables need not be repeated in the text.

-You unexpectedly find less transplant failure (ie graft failure or death) at 30 days for week-end LT (p12 line 13), even though these patients' were sicker (p11 line 29), and this held true in both unadjusted and adjusted models. At the same time, you find similar unadjusted (death-censored) graft failure rates at weekday and weekend (p12 line 13). These are very interesting findings, and I appreciate you attempt to find explanations. Further discussion on this issue is beyond the scope of a statistical review, but you may consider including a separate analysis of patient death.

Due to the fact that we did not observe any significant differences between day and night transplantation or organ procurement, we have given this less prominence in the manuscript than the day of the week assessment. We selected a narrative presentation of the major effects of day and night transplantation to avoid cluttering the manuscript with more tables. We chose this to ensure that the reader can see the true frequency of day and night transplantation and the impact this has on outcomes. We do not repeat any data in the main text that is tabulated in supplementary tables A4 and A5 but make reference to them if readers wish to see the hazard ratios for night time transplantation from the adjusted model. To avoid overcomplicating the manuscript we have not included additional tables, and do not feel that the day-night analysis warrants this, in light of the caveats and complexity in interpretation which render it liable to over interpretation (see detailed response to reviewer 3, point 2).

8. Minor comments:

-The many analyses incorporated into this manuscript make it a bit hard to read. Consider focusing the text on the major aetiological analysis (HRs), event probabilities could be presented in tables beneath the survival curves.

-Table A2 showing the full results for all risk factors is not explained, - HRs in bold seem to stem from the multivariable model, and the non-highlighted are then probably univariate? Consider excluding this table.

We have rendered the text more readable through making amendment in the light of all the reviewers' comments. We consider Table A2 to provide useful additional information and so have retained it as a supplementary table. We have clarified its relevance in the results section of the manuscript.

Reviewer: 4

The manuscript deals with "out-of-hours" clinical practice (weekend vs. weekdays; night-time vs. day-time), on liver transplantation (success vs. failure). Authors used a very relevant archive sample.

Authors used a very relevant archive sample.

Statistical procedures are suitable and can answer research questions.

I only have some questions about data presentation: Authors state there is not increased risk of failure with weekend or night-time surgical intervention. The unique exception is in the 30-days interval, where a higher failure is observed in weekday's interventions. But if we observe table 2, adjusted models are also significant for one year ($p=0.01$) and three years ($p=0.02$).

We would like to thank the reviewer for their consideration and positive comments regarding our manuscript. We were surprised when we identified a statistically significant reduced hazard of adverse outcomes for patients undergoing transplantation at the weekend compared to weekdays, in both the unadjusted and adjusted models. As we are unable to demonstrate a conclusive mechanism for any improvement in outcomes, and the analysis relies on statistical modelling that cannot account for many confounders (unmeasurable and potentially unrecognised) we felt that we could be confident that at least there was no additional risk in these off-hour periods (please also see response to reviewer 2, point 3). Any putative improved outcomes in out-of-hours liver transplant should be interpreted with caution, as this is the first description of an inverse weekend effect and due to the retrospective nature and large number of subjects, there is a risk of falsely identifying an association. As such we focus in the discussion that at least the weekend period is of no greater risk than weekdays and we explore the possibility of a benefit at weekends. We hope this finding stimulates discussion in the community and further attention to the role and impact of staffing and competing service needs through the week on critical patient outcomes.

There is a debate about p values in clinical data (there are proposals for p values lower than .005 for statistical significance; i.e. <http://dx.doi.org/10.1001/jama.2016.2152>; <http://dx.doi.org/10.1001/jama.2018.1536>). If authors agree with that proposal, I think it will be useful to declare it in data analysis subsection (and applied it to the rest of analyses).

Many thanks for highlighting this important and contentious area of discussion. We appreciate the importance of the open and accurate reporting and interpretation of statistical outcomes. We have attempted to be cautious in our interpretation of statistical findings and stated hazard ratios, confidence intervals and p-values for our major findings. We feel that this will allow the reader to interpret the data from both the uncorrected and corrected models themselves. Whilst we discuss the possibility that there was a significant association of improved outcomes with weekend liver transplantation, we have couched this in terms of a potential association due to the uncertainty of mechanism and the potential for confounders. We have avoided incorrectly using a $p<0.05$ as a marker of clinical importance or proof that the null hypothesis is false.

Also, the authors find several differences between patients with success / failure in liver transplantation. I think manuscript can provide relevant information about factors associated to failure / success liver transplantation, if auxiliary other data analyzes are performed. I. e., Logistic regression on significant variables (better, mixed linear analysis, because of several data are nested). In any case, the manuscript is sufficiently relevant as it is.

Once again we would like to thank the reviewer for this insight. The data set within the UK transplant registry is highly suited to explore a range of factors which may influence success and failure in liver transplantation. This is an area that has and continues to attract significant attention. For ease of interpretation and clarity of the manuscript we have avoided including any further modelling of other predictors of transplant outcome at this point. Specifically, we have focused on time to event modelling, rather than using logistic regression to model other outcomes, such as event probabilities. However, we fully agree that analysis of different endpoints could provide information about important areas of clinical interest but regard this as out of scope.

VERSION 2 – REVIEW

REVIEWER	Dag Olav Dahle Department of Transplantation, Oslo University Hospital, Rikshospitalet
REVIEW RETURNED	19-Nov-2018

GENERAL COMMENTS	I have no further concerns
----------------------------

REVIEWER	Wenceslao Peñate Universidad de La Laguna Tenerife, Spain
REVIEW RETURNED	20-Nov-2018

GENERAL COMMENTS	I think some answers from the authors are questionable, but as a scientific debate. I consider their answers are satisfactory and the study can be accepted
---